# A Closer Look at Personalized Fine-Tuning in Heterogeneous Federated Learning

## Abstract

Federated Learning (FL) enables decentralized, privacy-preserving model training but struggles to balance global generalization and local personalization due to non-identical data distributions across clients. Personalized Fine-Tuning (PFT), a popular post-hoc solution, fine-tunes the final global model locally but often overfits to skewed client distributions or fails under domain shifts. We propose adapting Linear Probing followed by full Fine-Tuning (LP-FT)—a principled centralized strategy for alleviating feature distortion [27]—to the FL setting. Through systematic evaluation across seven datasets and six PFT variants, we demonstrate LP-FT's superiority in balancing personalization and generalization. Our analysis uncovers federated feature distortion, a phenomenon where local fine-tuning destabilizes globally learned features, and theoretically characterizes how LP-FT mitigates this via phased parameter updates. We further establish conditions (e.g., partial feature overlap, covariate-concept shift) under which LP-FT outperforms fine-tuning, offering actionable guidelines for deploying robust FL personalization.

## 1 Introduction

Federated Learning (FL) [37] enables collaborative learning from decentralized data while preserving privacy, typically by training a shared global model, referred to as General FL (GFL). However, variations in client data distributions often limit GFL's effectiveness. Personalized FL (PFL) [24] addresses this by customizing models to individual clients. *Personalized Fine-Tuning* (PFT) [48], a simple and practical strategy in the PFL family, is a widely adopted post-hoc, plug-and-play approach to diverse GFL methods. As shown in Fig. 1(a), PFT fine-tunes the final global model from GFL to personalize it. This simple strategy ensures easy implementation and adaptation across FL scenarios.

Unlike *process-integrated PFL* methods (*e.g.*, those involving server-client coordination that modifies the entire federated training process [7, 5] or local training strategies that require iterative server feedback [25, 42]), PFT eliminates the need for costly global-training-dependent adaptations. Instead, it fine-tunes the final GFL model once post-training, ensuring simplicity, broad compatibility, and deployment robustness without redesigning the GFL framework (see Tab. 1). These characteristics establish PFT as a critical fallback strategy when process-integrated PFL approaches prove infeasible — particularly in scenarios where global training protocols are unmodifiable due to infrastructure lock-in or legacy FL infrastructure, or strict coordination agreement constraints (*e.g.*, healthcare systems bound by long-term service agreements). However, PFT often causes models to overfit on local data, thereby compromising the generalization of FL. This is particularly concerning in critical real-world applications, such as FL across multiple hospitals for disease diagnosis, where a local model must not only perform well on hospital patient data, but also generalize effectively to diverse patient populations that may be encountered on-site in the future [49]. Therefore, balancing

Table 1: Comparisons of Process-Integrated PFL vs. Post-Hoc PFT

| Criterion | Process-Integrated PFL | PFT (Post-Hoc) |
|---|---|---|
| **Global training modification** | **Required** (aggregation changes or iterative local training with server feedback) | **None** (algorithm-agnostic) |
| **Implementation Complexity** | **High** (client-server coordination, custom aggregation/regularization) | **Low** (single fine-tuning step, client autonomy, plug-and-play) |
| **Compatibility with GFL** | **Limited** (framework-specific) | **Broad** (process-agnostic) |

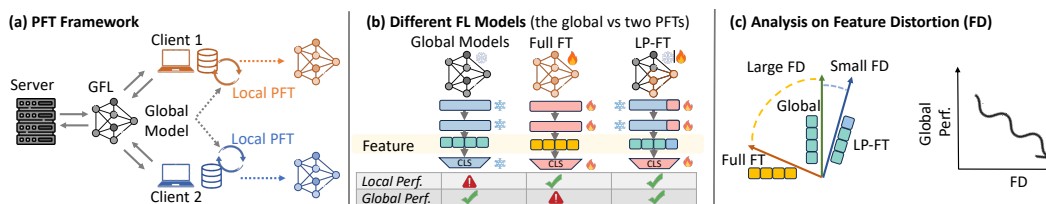

Figure 1: Overview of the problem setting and FL strategies investigated in this paper. (a) PFT framework, where each client fine-tunes a global model trained via GFL (e.g., FedAvg in this paper). Unlike process-integrated PFL, PFT focuses solely on the final fine-tuning stage with no further communication. (b) Three different FL models: the global FL model, the full-parameter FT (full FT) model, and the LP-FT model; their parameter updating patterns and local/global performance (perf.) under data heterogeneity; The fire icon indicates the actively tuned parameter, the frozen icon represents the fixed weight, and the mixed fire-frozen icon denotes the weight that is not actively tuned. (c) Visualization of feature distortion under PFL and its possible link to global generalization.

the optimization of individual client performance (personalization) with strong global performance (generalization across all clients) is crucial [48, 17].

In this work, we conduct a comprehensive evaluation of various strategies for PFT in heterogeneous FL environments under different distribution shifts, categorized as covariate shift [39, 13] and concept shift [20]. Despite meticulously tuning the hyper-parameters in some FT methods (full parameter FT, sparse FT [28] and Proximal FT [31]) adapted in FL, we observe persistent issues of local overfitting when increasing the local fine-tuning epochs, wherein localized performance gains are achieved at the significant cost of global generalization.

LP-FT [27]—a two-phase fine-tuning strategy that first updates *only* the linear classifier (Linear Probing, LP) before optimizing all parameters (Full Fine-Tuning, FT)—has demonstrated state-of-the-art performance in centralized learning by mitigating overfitting and enhancing domain adaptation. However, its potential to address FL challenges, such as client data heterogeneity and instability during decentralized personalization, remains unexplored. In FL, local fine-tuning risks overfitting to client distributions and diverging from globally useful representations. LP-FT's structured separation of updating the head and then fine-tuning offers a principled framework to stabilize personalization in non-IID settings while preserving global knowledge.

Yet, no work has rigorously evaluated LP-FT's efficacy in FL—a critical oversight given the growing demand for lightweight, flexible, and robust personalization strategies. Empirically, we conduct a comprehensive evaluation across seven datasets and diverse distribution shifts, benchmarking our adapted LP-FT against other advanced fine-tuning methods in our PFT framework. Our findings reveal two key insights: (1) existing PFT methods suffer from personalized overfitting, where local fine-tuning distorts feature representations, degrading global performance (Fig. 2); (2) LP-FT mitigates this issue, preserving generalization while enhancing local adaptation under extreme data heterogeneity. Further, extensive ablation studies (Fig. 4) confirm that LP-FT reduces federated feature distortion, establishing it as a strong and scalable baseline for PFT in FL.

Theoretically, we revisit *feature distortion*–a key challenge previously defined in centralized LP-FT as feature shifts under out-of-domain fine-tuning—in FL's unique setting of partially overlapping local and global distributions. Unlike centralized analyses [27], which assume a single ground-truth function, FL involves multiple client-specific ground-truth functions, necessitating a new theoretical framework. We address this by decoupling the feature extractor and classifier to analyze LP-FT's adaptation to heterogeneous client data. Further, we introduce a combined covariate-concept shift

setting, better reflecting real-world FL scenarios. Our analysis reveals conditions under which LP-FT outperforms full fine-tuning, advancing the understanding of fine-tuning strategies in FL.

This paper takes a closer look at PFT and establishes LP-FT as a theoretically grounded and empirically viable solution for FL's unique constraints. In summary, our contributions are threefold: (1) Methodologically, this paper presents the first systematic and in-depth study on the post-hoc and plug-and-play PFT framework and introduces LP-FT as an effective approach for handling diverse distribution shifts. We comprehensively demonstrate its ability to balance personalization and generalization in the FL setting. (2) Empirically, we conduct extensive experiments across seven datasets under various distribution shifts, complemented by thorough ablation studies. Our results validate the robustness of LP-FT and reveal overfitting tendencies in prior PFT methods. These empirical insights not only establish LP-FT as a strong baseline for PFT but also provide a foundation for future research in simple and flexible FL personalization. (3) Theoretically, we offer a rigorous theoretical analysis of LP-FT using two-layer linear networks, demonstrating its superior ability to preserve global performance compared to FT in both concept shift and combined concept-covariate shift scenarios.

## 2 Related Work

**Fine-Tuning** pre-trained models has gained prominence in centralized learning, particularly with the rise of foundation models [1]. However, fine-tuning with limited data often leads to overfitting. *Model soups* [47] and partial fine-tuning [29] further enhance adaptation by selectively updating model components. LP-FT [27], which combines linear probing with full fine-tuning, addresses feature distortions and provides insights into model adaptation under diverse shifts [43]. However, the effectiveness of these centralized fine-tuning strategies in the heterogeneous FL setting remains largely underexplored.

**Personalized FL** aims to address the challenges of decentralized learning with non-IID data. Classical *general FL (GFL)* methods, such as FedAvg [37], struggle in such settings. Despite the advancements in GFL methods (*e.g.*, FedNova [45]), FedProx [32], Scaffold [25]), their focus on building a single global model does not adequately address the data heterogeneity inherent in FL, leading to the emergence of *personalized FL (PFL)* [10, 50], which focuses on tailoring individualized models for each client. However, most PFL methods are *process-integrated*, requiring modifications to the global training pipeline through server-client coordination [7, 5] or iterative local training with server feedback [25, 42], or modifying training with customized clustering/regularization [11, 41]. These approaches impose constraints on flexibility and deployment, as we summarized in Tab. 1. In contrast, *post-hoc personalized fine-tuning (PFT)* [48] fine-tunes the final global model from GFL without modifying the training process, providing a lightweight and flexible approach for FL personalization. However, its potential is underexplored, possibly due to overfitting risks on client data. Additional discussion on personalization and fine-tuning is in App. B.

## 3 Empirical Study of PFT

To systematically investigate the challenges and opportunities in PFT, we present a comprehensive empirical study. First, in Sec. 3.1, we formalize the problem of PFT and characterize the spectrum of data heterogeneity to be studied. Next, Sec. 3.2 details our experimental setup, including datasets and PFT strategies under consideration. Our investigation then addresses a critical yet understudied phenomenon: Sec. 3.3 analyzes the prevalence of *personalized overfitting* in PFT across distribution shifts, even with careful hyper-parameter tuning. Motivated by this finding, Sec. 3.4 introduces LP-FT and benchmarks its performance against alternative PFT strategies in FL, showing its superior ability to balance local adaptation with global knowledge retention. Finally, to uncover the mechanistic drivers of generalization challenges, Sec. 3.5 conducts the first systematic analysis of federated feature distortion—quantifying how client-specific fine-tuning trajectories alter latent representations and degrade model robustness.

### 3.1 Overview and Definitions

**Problem Setting.** In a FL setting, each client $i \in [C]$ has a local dataset $(\mathbf{X}_i, \mathbf{Y}_i)$ generated from a potentially distinct distribution, which may differ across clients due to distribution shifts. PFT

aims to optimize local model parameters $\theta_L$ for each client, initialized from a well-trained global model $\theta_G$. The objective is to minimize the local loss $\mathcal{L}_L(\theta_L)$ for improved local performance while ensuring that the global loss $\mathcal{L}_G(\theta_L)$ remains close to that of a pre-trained global model. This creates a trade-off between personalization (minimizing local loss) and maintaining generalization (minimizing global loss) across clients. The global data distribution $\mathcal{D}_G$ is defined as a mixture of the local distributions $\mathcal{D}_i$, given by $\mathcal{D}_G = \frac{1}{C} \sum_{i \in [C]} \mathcal{D}_i$.

We formally define distributions of interests, concept shift and covariate shift that directly lead to feature shift in heterogeneous FL context[1], following [33].

**Covariate Shift** refers to variations in the input feature distribution across clients while keeping the conditional distribution of the output given the input consistent. Formally, for any pair of clients $i$ and $j$ with $i \neq j$, the data-generating process is characterized by:

$$P_i(x) \neq P_j(x), \;\; \text{but} \;\; P_i(y \mid x) = P_j(y \mid x) \;\; \text{for all } i \neq j.$$

This means that while clients $i$ and $j$ may have different input distributions $P_i(x)$ and $P_j(x)$, the conditional distribution $P(y \mid x)$ remains consistent across all clients.

**Concept Shift** occurs when the conditional relationship between input features and outputs differs across clients, while the input feature distribution remains unchanged. Formally, for any two clients $i$ and $j$ with $i \neq j$, the data-generating process satisfies:

$$P_i(y \mid x) \neq P_j(y \mid x), \;\; \text{but} \;\; P_i(x) = P_j(x) \;\; \text{for all } i \neq j.$$

This implies that although all clients share the same input distribution $P(x)$, the conditional distribution $P_i(y \mid x)$ varies, reflecting different mappings between features and labels across clients.

## 3.2 Empirical Analysis Settings

**Datasets with Covariate Shift.** We include `Digit5`, `DomainNet`, `CIFAR10-C`, and `CIFAR100-C`. `Digit5` and `DomainNet` belong to the *feature-shift* subgroup, where the data features represent different subpopulations within the same classes. For example, `Digit5` contains 10-digit images collected from various sources with different backgrounds, such as black-and-white for MNIST and colorful digits for synthetic datasets. `CIFAR10-C` and `CIFAR100-C` fall under the *input-level shift* category, where 50 types of image corruptions are introduced for evaluation. We simulate 50 clients, each corresponding to a specific corruption type, as detailed in previous works [12, 38, 4]. A detailed explanation of the data splitting and its introduction is provided in Tab. 4 in Appendix. The visualizations of data are provided in Fig. 5.

**Datasets with Concept Shift.** `CheXpert` and `CelebA` are included for this part, whereas both belong to the *spurious correlation-based shift* subgroup, which involves misleading relationships in the training data that models may exploit, despite being unrelated to the actual target. This reliance can lead to poor performance when such correlations are absent in new data, classifying it as a form of concept shift [20]. Similarly, Tab. 4 and Fig. 5 provide further details.

**Fine-tuning Strategies.** Our study focuses on post-hoc PFT, a plug-and-play framework that operates exclusively after GFL training. Unlike conventional fine-tuning in centralized settings that primarily addresses domain adaptation by transferring a model from a source to a disjoint target domain, PFT operates on a global model pre-trained via GFL, which has already been exposed to heterogeneous client data during collaborative training and must balance local performance (adapting to a client's unique distribution) with global performance (avoiding overfitting to statistically biased local updates and preserving cross-client generalizability).

In this study, we establish a suite of fine-tuning strategies that can be easily integrated into PFL as **baselines** for PFT: *Full-parameter FT* is a naive FT strategy. It adjusts all model parameters. *Proximal FT* [31] aims to preserve the pre-trained model's original knowledge. It applies proximal regularization to penalize large deviations from the initial model parameters, helping to maintain generalization. *Sparse FT* [28] promotes sparsity in parameter updates. It adjusts only the most relevant weights, enhancing efficiency while regularizing the training from overfitting. *Soup FT* [46] improves robustness by averaging the weights of multiple fine-tuned model instances. Each instance is trained with different initializations, creating a "model soup" that integrates their strengths. *LSS FT* [3]

---

[1]We also realized that LP-FT can be effective for label shift settings as the results shown in App. D.2.

(Local Superior Soups) is an innovative model interpolation-based local training technique designed to enhance FL generalization and communication efficiency by encouraging the exploration of a connected low-loss basin through optimizable and regularized model interpolation. Each strategy is designed to balance model performance with different priorities, such as preserving knowledge, enhancing robustness, or improving efficiency. A more detailed experiment setting is presented in App. C.[2]

## 3.3 Global and Local Performance Trends in PFT Baselines

In practice, PFT is susceptible to overfitting to local data, due to the relatively small amount of data available at local clients. Note that the *overfitting* defined in the FL context is characterized by *a consistent improvement in local performance while global performance noticeably deteriorates [48, 2] – the average gain in local performance can be smaller than the loss in global performance.* To measure the model's overall local and global performance, we measure the averaged client-wise local and global accuracy. Specifically, this metric reflects the average performance between clients' *local* test accuracy and their local model's accuracy on the rest of the clients (*global* accuracy). The metric's decreasing trend with increasing local training epochs during the finetuning stage indicates personalized overfitting. Notably, this trend persists even when considering only global performance metrics, as local performance tends to show increases in PFT under overfitting conditions.

In all subplots of Fig. 2, we evaluate baseline PFT strategies under diverse distribution shifts, including input-level shifts (`CIFAR100-C`), feature-level shifts (`Digit5`), and spurious correlation-based shifts (`CheXpert`). We systematically adjusted hyperparameters to evaluate their impact on performance. Fig. 2a demonstrates that overfitting persists even when fine-tuning with reduced learning rates. Fig. 2b reveals that gradient sparsity adjustments (where higher sparsity rates mask more parameter updates) fail to mitigate overfitting as training epochs increase. Fig. 2c further shows that proximal regularization terms, designed to bias updates toward the initial global model, still exhibit global performance decay despite regularization.

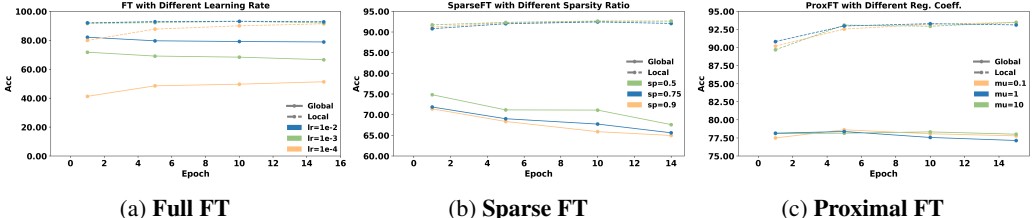

|  (a) **Full FT** | (b) **Sparse FT** | (c) **Proximal FT** |

Figure 2: Visualization of the prevalence of personalization overfitting across different distribution shift scenarios, where (a) shows the global and local accuracy under different learning rates for full-parameter fine-tune; (b) shows the different sparsity rate for sparse fine-tune; (c) shows the different regularization strength under the proximal fine-tune. In all subfigures, the global accuracy is shown as the solid line, and the local accuracy is shown as the dashed line. As shown, global accuracy consistently declines while local accuracy either increases or remains stable across different hyperparameter settings. This suggests that PFT baseline methods are prone to overfitting, even with careful hyperparameter tuning.

## 3.4 Performance Comparison

**Linear Probing then Fine-Tuning.** To address the challenge of personalized overfitting in conventional fine-tuning methods within PFT, we propose a simple yet effective approach through Linear Probing followed by Fine-Tuning (LP-FT) for FL. The idea is motivated by LP-FT [27]—a two-phase fine-tuning strategy in centralized training that first updates *only* the linear classifier (Linear Probing, LP) before optimizing all parameters (Full Fine-Tuning, FT) to improve out-of-domain performance while preserving in-domain performance. We adapt the strategy in PFT as follows: *In practice, clients initialize weights from the model after GFL, first perform linear probing, and then fine-tune the full model as shown in Fig. 1 (b).* This LP-FT approach achieves strong personalization while maintaining generalizability across diverse clients.

---

[2]We primarily focus on CNN-based models. We also include parameter-efficient fine-tuning results on transformer-based models in Appendix Tab. 5.

Table 2: Performance of various PFT strategies. **Red** represents the *input shift* subgroup; green from the *feature-shift* subgroup; blue the *spurious correlation-based shift* subgroup. Each experiment is performed three times independently with different random seeds, and the standard deviation of the results is presented in parentheses. ↑ indicates that higher values are better, while ↓ indicates that lower values are better.

| Dataset | Method | Local ↑ | Global ↑ | C-Std. ↓ | Worst ↑ | Average ↑ |
|---|---|---|---|---|---|---|
| **CIFAR10-C** | FT | 54.50 (0.64) | 44.16 (0.13) | **10.04 (0.06)** | 19.83 (0.18) | 39.50 (0.33) |
| | Proximal FT | 61.76 (0.13) | 53.58 (0.14) | 11.61 (0.08) | 25.82 (0.12) | 47.05 (0.07) |
| | Soup FT | 56.36 (0.23) | 44.94 (0.06) | 10.22 (0.06) | 20.47 (0.35) | 40.59 (0.09) |
| | Sparse FT | 61.31 (0.01) | 50.21 (0.17) | 11.10 (0.11) | 24.56 (0.09) | 45.36 (0.04) |
| | LSS FT | 56.21 (0.33) | 46.81 (0.04) | 10.05 (0.08) | 21.61 (0.37) | 43.67 (0.08) |
| | LP-FT | **63.55 (0.04)** | **55.35 (0.01)** | 12.45 (0.01) | **26.33 (0.06)** | **48.41 (0.03)** |
| **CIFAR100-C** | FT | 20.05 (0.05) | 14.45 (0.04) | **5.37 (0.02)** | 3.37 (0.06) | 12.62 (0.03) |
| | Proximal FT | 27.38 (0.15) | 19.96 (0.11) | 6.90 (0.04) | 4.84 (0.04) | 17.41 (0.05) |
| | Soup FT | 20.99 (0.24) | 14.81 (0.04) | 5.48 (0.03) | 3.56 (0.01) | 13.12 (0.06) |
| | Sparse FT | 28.93 (0.04) | 20.66 (0.02) | 7.75 (0.02) | 5.05 (0.09) | 18.15 (0.10) |
| | LSS FT | 20.54 (0.19) | 15.42 (0.03) | 5.32 (0.03) | 3.62 (0.01) | 14.22 (0.06) |
| | LP-FT | **32.60 (0.14)** | **25.44 (0.10)** | 9.66 (0.04) | **5.92 (0.06)** | **21.32 (0.04)** |
| Digit5 | FT | 91.17 (0.90) | 67.87 (0.74) | 22.93 (0.28) | 42.03 (0.48) | 67.02 (0.70) |
| | Proximal FT | **92.09 (0.18)** | 81.40 (0.03) | 15.04 (0.15) | 61.71 (0.16) | 78.40 (0.09) |
| | Soup FT | 91.82 (0.34) | 70.82 (0.43) | 21.99 (0.67) | 45.10 (1.27) | 69.02 (0.65) |
| | Sparse FT | 91.43 (0.31) | 76.89 (0.72) | 17.90 (0.38) | 54.21 (0.56) | 74.21 (0.35) |
| | LSS FT | 91.59 (0.28) | 73.13 (0.30) | 22.04 (0.53) | 45.32 (1.13) | 71.15 (0.53) |
| | LP-FT | 91.20 (0.04) | **82.78 (0.05)** | **13.75 (0.02)** | **65.80 (0.02)** | **79.92 (0.02)** |
| DomainNet | FT | 64.90 (1.18) | 42.48 (0.58) | 17.49 (0.75) | 22.31 (0.93) | 43.23 (0.52) |
| | Proximal FT | 67.20 (1.39) | 56.05 (0.27) | **16.68 (0.36)** | 33.20 (1.79) | 52.60 (0.35) |
| | Soup FT | 67.48 (0.61) | 44.27 (0.46) | 18.44 (0.42) | 23.73 (1.24) | 44.49 (0.54) |
| | Sparse FT | **69.62 (0.53)** | 50.24 (0.44) | 18.14 (0.17) | 27.89 (0.15) | 49.14 (0.45) |
| | LSS FT | 66.37 (0.53) | 45.34 (0.40) | 18.02 (0.38) | 22.63 (1.05) | 45.75 (0.42) |
| | LP-FT | 68.50 (0.19) | **57.52 (0.20)** | 17.36 (0.21) | **34.53 (0.44)** | **53.52 (0.19)** |
| **CheXpert** | FT | 76.18 (0.41) | 76.25 (0.56) | 0.35 (0.13) | 76.31 (0.76) | 76.25 (0.44) |
| | Proximal FT | 76.44 (0.07) | 76.63 (0.09) | 0.71 (0.09) | 76.81 (0.07) | 76.63 (0.07) |
| | Soup FT | 77.51 (0.15) | 77.49 (0.31) | 0.48 (0.07) | 77.46 (0.43) | 77.49 (0.26) |
| | Sparse FT | 77.29 (0.13) | 77.20 (0.14) | **0.31 (0.11)** | 77.11 (0.25) | 77.20 (0.14) |
| | LSS FT | 77.49 (0.14) | 77.51 (0.28) | 0.52 (0.08) | **77.53 (0.37)** | 77.52 (0.24) |
| | LP-FT | **77.64 (0.37)** | **77.54 (0.37)** | 0.53 (0.41) | 77.43 (0.71) | **77.54 (0.37)** |
| **CelebA** | FT | 90.55 (1.20) | 73.76 (2.15) | 18.79 (3.64) | 53.52 (5.51) | 72.39 (2.84) |
| | Proximal FT | **93.74 (0.59)** | 81.11 (0.82) | 13.39 (1.14) | 67.50 (2.10) | 80.78 (0.90) |
| | Soup FT | 89.42 (2.16) | 75.28 (1.11) | 16.29 (1.19) | 57.79 (2.90) | 74.17 (1.50) |
| | Sparse FT | 91.43 (0.48) | 77.32 (1.46) | 14.16 (2.57) | 62.94 (4.34) | 77.65 (1.65) |
| | LSS FT | 89.17 (2.05) | 77.35 (1.03) | 16.23 (1.28) | 59.64 (2.86) | 76.74 (1.46) |
| | LP-FT | 93.24 (0.17) | **83.32 (0.31)** | **11.18 (0.14)** | **71.89 (0.75)** | **82.82 (0.64)** |

**Experimental Settings.** To isolate the impact of PFT strategies and avoid conflating gains from GFL optimization, we standardize the GFL stage by fixing its method to FedAvg, the foundational and most widely used GFL method. Within this framework, we focus on comparing different *post-hoc* FT methods to demonstrate the effectiveness of LP-FT in PFT (see Fig. 1 (a)). After the GFL stage, all the clients further fine-tune the obtained global model using local data for 15 epochs for personalization. The final models are evaluated using the *metrics* described below. Details of the datasets, preprocessing steps, data splitting, and models used are provided in App. C.3, Tab. 4.

**Metrics.** We adapt five metrics in our baseline experiments: *(1) Local Accuracy (Local)* measures the performance of the PFT model on the client's local test set. Higher *Local Acc* indicates better personalization. *(2) Global Accuracy (Global)* measures the PFT model's average test accuracy over all other clients' test sets. Higher *Global Acc* indicates better generalization. *(3) Client-wise Standard Deviation (C-Std.)* calculates the standard deviation of local test accuracies across all clients. Lower *C-Std.* indicates less variance in performance among clients. *(4) Worst Accuracy (Worst)* reports the lowest test accuracy among all clients. The closer this value is to *Local Acc*, the better the worst-case generalization. *(5) Average* reports the average of both *Local Acc* and *Global Acc*, providing a better

understanding of the tradeoff between personalization (local performance) and generalization (global performance). All metrics, except *C-Std.*, are averaged over the number of clients, and higher values are preferable. For *C-Std.*, lower values are better.

**Results.** Our results are presented in Tab. 2, where the best method is highlighted in **bold**. Datasets with the same distribution shift pattern are grouped into the same colors as detailed in the caption. Tab. 2 shows that LP-FT consistently achieves the highest global and average accuracy across most datasets, demonstrating strong generalization and personalization performances, particularly in challenging conditions like `CIFAR100-C` and `CIFAR10-C`. Sparse FT also performs well, especially in `Digits5` and `DomainNet`, but generally lags behind LP-FT. LSS FT, Soup FT and Proximal FT show mixed results, with stronger performance in specific datasets such as `CheXpert` but weaker overall compared to LP-FT. Standard fine-tuning consistently underperforms, highlighting the limitations of basic fine-tuning methods in heterogeneous data scenarios.

## 3.5 Insight and Explanation on the Observations

Given the unique design of LP-FT, we hypothesize that its superior performance in PFT stems from its ability to mitigate federated feature distortion — a phenomenon where client-specific fine-tuning disrupts the global model's learned representations. We empirically validate this hypothesis through a systematic analysis of feature space dynamics across diverse data heterogeneity scenarios.

**Federated Feature Distortion.** Consider a feature extraction function $f : \mathcal{X} \rightarrow \mathbb{R}^k$, which maps inputs from the input space $\mathcal{X}$ to a representation space $\mathbb{R}^k$. Let $\theta_G$ denote the global pre-trained model and $\theta_i$ the fine-tuned model after local fine-tuning for client $i$. Assume there are $C$ clients in total, each with $n$ samples. Let $x_{c,j}$ represent the $j$-th data point of the $c$-th client. The *federated feature distortion* $\Delta_c(f)$ quantifies the change in features after fine-tuning for the $c$-th client, defined as the average $\ell_2$ distance between the representations produced by the global model and the locally fine-tuned model over all data points across all clients. Formally, it is expressed as: $\Delta_c(f) = \frac{1}{n} \sum_{j=1}^{n} \| f(\theta_G; x_{c,j}) - f(\theta_c; x_{c,j}) \|_2$, where $\| \cdot \|_2$ is the $\ell_2$ distance in the representation space $\mathbb{R}^k$. We compute the average of $\Delta_c(f)$ across all clients to represent the feature distortion in the PFT setting, as shown in Fig. 3.

**Empirical Validation.** To quantify federated feature distortion, we measure the $\ell_2$ distance between global and locally fine-tuned feature representations using `DomainNet` and `Digit5`. As shown in Fig. 3(a), the full FT method induces severe feature distortion, correlating with a significant drop in global accuracy, whereas LP-FT maintains stable global performance with lower distortion.

To further isolate the effect of feature distortion from local loss magnitude, we apply loss flooding [19] to control local training loss levels (0.1, 0.5, 1.0). Fig. 3(b) shows that at fixed local loss levels, LP-FT consistently outperforms FT in global accuracy, confirming that its advantage stems from reduced feature distortion rather than differences in local optimization dynamics.

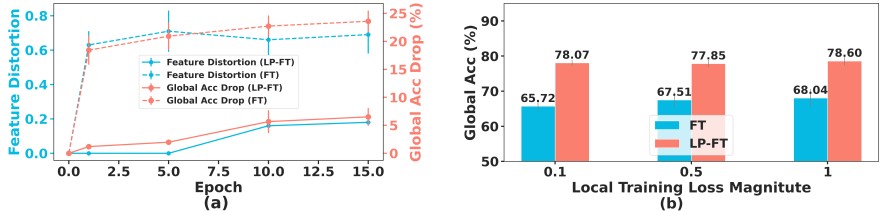

Figure 3: Observations of the feature distortion in PFT setting, where (a) presents the positive correlation between global performance drops and feature distortion intensity on `DomainNet` and (b) presents the ablation study on preserving federated features with controlled local train loss on `Digit5`. We set local loss thresholds (0.1, 0.5, and 1.0) and used gradient ascent when the loss fell below, ensuring training loss fluctuated around these points.

## 4 Theoretical Analysis of the LP-FT in FL

Building on our empirical observations in Sec. 3, where LP-FT consistently outperforms baseline PFT methods and demonstrates a significant reduction in federated feature distortion, we now present

a theoretical analysis to uncover the mechanistic principles underlying its success. To understand how feature learning impacts generalization error in PFT, we decompose the data-generating function and the model into two components: a feature extractor and a linear head. This decomposition allows us to distinguish between the learned features and their influence on performance. Specifically, in Sec. 4.1 and Sec. 4.2, we formalize concept and covariate shifts within a two-layer linear network and examine how LP-FT effectively adapts to these shifts, outperforming full-parameter fine-tuning (FT) in FL.

**Overview of Theoretical Analysis:** To compare the performance of LP-FT and FT, we make assumptions about the data-generating function for clients (Assumption 4.1) and a specific model structure (Assumption 4.2). Based on these assumptions, we analyze the global performance of LP-FT and FT under concept shift (Theorem 4.4) and combined concept-covariate shift (Theorem 4.5).

## 4.1 LP-FT's Global Performance Under Concept Shift

In this section, we analyze LP-FT's performance compared to FT under concept shift. To facilitate a rigorous theoretical study, we define the data-generating process and model structure across clients, assuming both are represented by two-layer linear networks, as in [27].

**Assumption 4.1** (Data-Generating Process). The data-generating function for client $i$ is given by $y_i = V_i^{*T} B_* x_i$ for all $i \in [C]$, where $y_i \in \mathbb{R}$, $C$ is the number of clients, $x_i \in \mathbb{R}^d$, $B_* \in \mathbb{R}^{k \times d}$, and $V_i^* \in \mathbb{R}^k$. All clients share a common feature extractor $B_*$, assumed to have orthonormal rows, while their linear heads $V_i^*$ differ. Each $V_i^*$ decomposes as $V_i^* = \begin{bmatrix} V_{com}^{*T} & \lambda e_i^T \end{bmatrix}^T$, where $V_{com}^* \in \mathbb{R}^m$ is shared across clients, $e_i \in \mathbb{R}^C$ is a unit vector, and $\lambda$ controls heterogeneity. Here, $m + C = k$.

This assumption distinguishes between a shared and client-specific component in the data-generating functions, allowing analysis of both global and local performance of PFT methods after fine-tuning.

**Assumption 4.2** (Model Structure). The training model is a two-layer linear network defined as $y = V^T B x$, where $V \in \mathbb{R}^k$ is the linear head and $B \in \mathbb{R}^{k \times d}$ is the feature extractor. The dimensions of $V$ and $B$ match Assumption 4.1, allowing the model to learn both shared and client-specific data components.

In PFT settings, our objective is to evaluate the performance of a model on both global and local data. By local data, we refer to the data of a specific client undergoing fine-tuning (e.g., client $i$). The local and global losses are defined using the Mean Squared Error (MSE) as follows:

$$\mathcal{L}_L(V, B) = \mathbb{E}_{(x,y) \sim \mathcal{D}_i} \left[ \frac{1}{2} (V^T B x - y)^2 \right] = \mathbb{E}_{x \sim \mathcal{D}_i} \left[ \frac{1}{2} (V^T B x - V_i^{*T} B_* x)^2 \right],$$

$$\mathcal{L}_G(V, B) = \mathbb{E}_{(x,y) \sim \mathcal{D}_G} \left[ \frac{1}{2} (V^T B x - y)^2 \right] = \frac{1}{C} \sum_{i \in [C]} \mathbb{E}_{x \sim \mathcal{D}_i} \left[ \frac{1}{2} (V^T B x - V_i^{*T} B_* x)^2 \right].$$

Since this section focuses on concept shift, we assume all clients' data is drawn from similar distributions. Accordingly, we assume for every client $i \in [C]$, the input features satisfy $\mathbb{E}_{x \sim \mathcal{D}_i}[xx^T] = I_d$.

With the theoretical framework established by Assumptions 4.1 and 4.2, we compare the global performance of LP-FT and FT, highlighting cases where LP-FT outperforms FT. As demonstrated in [6] FedAvg learns a shared data representation among clients if such a common representation exists. In a PFT setting, the initial model is trained on data from all clients to capture their shared components. Thus, we initialize the model parameters as $B_0 = B_*$ and $V_0 = \begin{bmatrix} V_{com}^{*T} & \mathbf{0} \end{bmatrix}^T$. In LP-FT, a step of linear probing first updates $V_0$ using local data while keeping $B_0$ fixed, followed by full fine-tuning to update both $V$ and $B$. In contrast, FT performs only the second step. The following lemma characterizes $B$ after one gradient descent step in FT, forming the basis for our comparison.

**Lemma 4.3.** *Under Assumptions 4.1 and 4.2, and assuming that $\mathbb{E}_{x \sim \mathcal{D}_i}[xx^T] = I_d$ for all clients $i \in [C]$, let the initial parameters before starting FT be $B_0 = B_*$ and $V_0 = \begin{bmatrix} V_{com}^{*T} & \mathbf{0} \end{bmatrix}^T$. Assume fine-tuning is performed locally on the data of the $i$-th client. Let $B_{FT}$ denote the feature extractor matrix after a single gradient descent step (processing the entire dataset once) with learning rate $\eta$. If $(b_j^{FT})^T$ is the $j$-th row of $B_{FT}$, then:*

$$\mathbb{E}\left[ (b_j^{FT})^T \right] = (b_j^*)^T + \eta \lambda (V_0)_j (b_{m+i}^*)^T,$$

*where $(b_j^*)^T$ is the $j$-th row of $B_*$, and $(V_0)_j$ is the $j$-th element of $V_0$ for $j \in [k]$.*

This lemma examines the impact of FT on the feature extractor $B_{FT}$, highlighting the deviations from the pre-trained matrix $B_0 = B_*$. Given that all clients share the same $B_*$ in their labeling functions, substantial changes to the feature extractor can degrade global performance. Since the matrix $B$ functions as the feature extractor in our framework, significant feature distortion occurs when $B_{FT}$ deviates considerably from $B_*$. Building on Lemma 4.3, Theorem 4.4 offers a comparative analysis of the global performance of LP-FT versus FT in the context of concept shift.

**Theorem 4.4.** *Under Assumptions 4.1 and 4.2, and assuming $\mathbb{E}_{x \sim \mathcal{D}_i}[xx^T] = I_d$ for all clients $i \in [C]$, let the initial model parameters be $B_0 = B_*$ and $V_0 = \begin{bmatrix} V_{com}^{*}{}^T & \mathbf{0} \end{bmatrix}^T$. Let $B_{FT}$ and $V_{FT}$ denote the parameters of the FT method after one gradient descent step (processing the entire dataset once). For LP-FT, let $B_{LPFT}$ and $V_{LPFT}$ denote the parameters after (i) linear probing, which optimizes $V$ with $B$ fixed at $B_*$, and (ii) one gradient descent step with learning rate $\eta$. Then:*

$$\mathcal{L}_G(V_{LPFT}, B_{LPFT}) \leq \mathcal{L}_G(V_{FT}, B_{FT}).$$

This theorem characterizes the global performance of LP-FT, suggesting that under concept shift, LP-FT achieves better performance on global data than FT. When starting from a model initialized to capture the shared feature extractor and linear head among clients, LP-FT is more effective in minimizing global loss, aligning with common FL scenarios where the initial model leverages shared client structure.

### 4.2  LP-FT's Global Performance under Combined Concept and Covariate Shifts

In the previous section, we assumed all clients' data came from the same distribution with $\mathbb{E}_{x \sim \mathcal{D}_i}[xx^T] = I_d$. However, this may not hold in many practical scenarios. To address this, we introduce covariate shift, where each client's data is generated as $x_i = e_i + \epsilon z$, with $z \sim \mathcal{N}(0, I)$, $e_i$ as a client-specific shift, and $\epsilon$ controlling the noise level. This extension captures the non-iid nature of data among clients and provides a framework to model data heterogeneity. The model structure and data-generating assumptions remain consistent with Sec. 4.1. This section thus considers both concept and covariate shifts. Theorem 4.5 analyzes the impact of heterogeneity on the global performance of LP-FT and FT.

**Theorem 4.5.** *Under Assumptions 4.1 and 4.2, let each client's data be $x_i = e_i + \epsilon z$, where $z \sim \mathcal{N}(0, I)$ and $e_i$ is a client-specific shift. Assume the initial parameters are $B_0 = B_*$ and $V_0 = \begin{bmatrix} V_{com}^{*}{}^T & \mathbf{0} \end{bmatrix}^T$. Let $B_{FT}, V_{FT}$ be the FT parameters after one gradient descent step, and $B_{LPFT}, V_{LPFT}$ be the LP-FT parameters after linear probing and one gradient descent step (with learning rate $\eta$). Then, there exists a threshold $\lambda^*$ such that for all $\lambda \leq \lambda^*$:*

$$\mathcal{L}_G(V_{LPFT}, B_{LPFT}) \leq \mathcal{L}_G(V_{FT}, B_{FT}).$$

*Remark* 4.6. In Theorem 4.5, the parameter $\lambda$ characterizes the level of heterogeneity among clients. The theorem shows that under both covariate and concept shifts, LP-FT outperforms FT in low heterogeneity settings ($\lambda \leq \lambda^*$), highlighting its advantage in maintaining generalization. To further reinforce the theoretical insights and cover more extensive settings, App. D.3 provides extensive empirical validation, confirming the global superiority of LP-FT over FT under combined concept-covariate shifts. While the theoretical analysis in Theorem 4.5 focuses on the low heterogeneity regime, the experiments in App. D.3 explore a broader range, including both high and low heterogeneity levels. Notably, LP-FT consistently outperforms FT across all heterogeneity regimes, aligning with our theoretical results in Sec. 4.2, particularly for deep neural networks in realistic PFT settings. These findings validate and extend our theoretical insights, demonstrating LP-FT's robustness and superiority in diverse distribution shift scenarios (see also App. F).

## 5  Conclusion

In this work, we studied an important PFL paradigm – PFT and tackled its key challenge of balancing local personalization and global generalization. We establish LP-FT as a theoretically grounded and empirically robust solution for PFT. Our work demonstrates that LP-FT effectively mitigates federated feature distortion, balancing client-specific adaptation with global generalization under extreme data heterogeneity. Methodologically, we are the first to adapt LP-FT to post-hoc PFT; empirically, we validate LP-FT's superiority across seven datasets; theoretically, we formalize its advantages in FL's unique covariate-concept shift regime. This work advances lightweight, deployable personalization for real-world FL systems.

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
