# OpenReview forum: "A Closer Look at Personalized Fine-Tuning in Heterogeneous Federated Learning"
_NeurIPS.cc/2025/Conference — Submitted to NeurIPS 2025_

### Official Review · Reviewer_PKSf · 2025-06-23

**Clarity:** 4
**Significance:** 3
**Originality:** 3
**Rating:** 5
**Confidence:** 4

**Summary:**

This paper proposes a two-stage fine-tuning method called LP-FT to address the trade-off between personalization and global generalization in the process of joint personalized fine-tuning. To this end, LP-FT first retains global features in the pre-trained model through linear probing, and then fully fine-tunes to improve local adaptation. This method provides clear theoretical support. It has been proven to outperform existing methods in performance through extensive experiments on multiple datasets and model settings.

**Questions:**

Theorem 4.5 states that "LP-FT outperforms FT when λ ≤ λ*", but the performance behavior under high heterogeneity (λ > λ*) is not covered by the theoretical analysis. What about the condition of λ > λ*?

**Ethical Concerns:**

["NO or VERY MINOR ethics concerns only"]

**Final Justification:**

I appreciate the authors' response. I would stay with the current rating.

**Limitations:**

yes

**Quality:**

3

**Strengths And Weaknesses:**

Strength:

1. Overall, this paper is well-written and easy to follow. The presentation of the Figures describing the motivation is well presented and easy to understand.

2. This paper focuses on the significant challenge of the balance of global generalization and local personalization, which is meaningful.

3. The theoretical analysis is solid and comprehensive.

4. Through extensive experimental results, this method is significantly better than the SOTA methods in personalized model fine-tuning tasks.


Weakness:

1. The organization of this work could be improved, as vanilla works for better understanding.

2. It could be better to compare with some personalized federated learning frameworks.

3. The results analyze the personalized and global accuracy. What about the fairness across the clients?

---

> ### Author Rebuttal · Authors · 2025-07-31
>
> We thank Reviewer PKSf for the very positive assessment.The acknowledgement of the paper’s **clear presentation**, **meaningful focus**, **comprehensive theory**, and **strong empirical gains** is greatly appreciated.
>
> The remaining points are minor clarifications, all addressed below without affecting our core contributions and claims.
>
>
> **1. Clarification on High-Heterogeneity Regime ($\lambda > \lambda^*$)**
>
> Our analysis in **Section&nbsp;4.2** focuses on combined concept + covariate shift under a *low-heterogeneity* regime. As shown in **Theorem 4.5**, LP-FT provably outperforms FT when $\lambda \le \lambda^{*}$. Bounding heterogeneity is standard in FL theory; when client distributions become completely disjoint, the global objective decomposes into a sum of unrelated losses, so a single model provides no benefit over training one model per client [1, 2]. For completeness, we empirically sweep a wide range of $\lambda$ in **Appendix F** to give a better understanding of our **Theorem 4.5**.
>
> [1] Li, Tian, et al. "Federated optimization in heterogeneous networks." Proceedings of Machine learning and systems 2 (2020): 429-450.
> [2] Karimireddy, Sai Praneeth, et al. "Scaffold: Stochastic controlled averaging for federated learning." International conference on machine learning. PMLR, 2020.
>
> **2. Response to Weaknesses W1–W3**
>
> **W1: Paper Organization.**
> Our **Section 1 Introduction** already lists the three core contributions (methodological, empirical, theoretical) and each main **Section 3, 4, 5** begins with a short roadmap to reinforce that structure. We will add brief sign-posting statements to further improve readability.
>
> **W2: Choice of Baselines.**
> Process-integrated PFL frameworks modify aggregation, add coordination, or embed new objectives—precisely the variables we hold constant. Including them in the main table would mix improvements from modified global training with those from post-hoc personalization, obscuring the effect we aim to isolate. Thus, Table 2 reports only post-hoc fine-tuning variants, all applied to the same FedAvg global model (stated in **Section 3.2 Line 151 to 157**). This setup eliminates confounding factors from global training, enabling a clean and fair assessment of post-hoc methods.
>
> **W3: Fairness Metrics.**
> Although fairness is orthogonal to our core goal of balancing local and global accuracy, **Table 2** already reports **client-wise standard deviation and worst-case accuracy**, metrics that directly quantify performance disparities across clients.

---

> > ### Comment · Reviewer_PKSf · 2025-08-05
> >
> > I appreciate the authors' response. I would stay with the current rating.

---

> > > ### Author Response · Authors · 2025-08-06
> > >
> > > Thank you for your kind feedback and for acknowledging our response in addressing your concerns. We truly appreciate your positive evaluation and thoughtful consideration.

---

### Official Review · Reviewer_FGp1 · 2025-07-01

**Clarity:** 2
**Significance:** 2
**Originality:** 2
**Rating:** 3
**Confidence:** 4

**Summary:**

This paper propose a method called LP-FT -- a two-phase strategy that first trains only the classifier head before unfreezing the backbone for full fine-tuning, to improve the out-of-domain performance while preserving in-domain performance.

**Questions:**

- Why choose these particular baselines?
- What is the pretrained model? What does it pretrained on and what is the model architecture.
- LP-FT doesn't outperform all baselines, can authors analyse more and provide more insights why LP-FT doesn't perform well on certain cases.

**Ethical Concerns:**

["NO or VERY MINOR ethics concerns only"]

**Final Justification:**

I have carefully reviewed the authors' responses and discussions with other reviewers and AC.

**Limitations:**

See above

**Paper Formatting Concerns:**

No particular concern.

**Quality:**

3

**Strengths And Weaknesses:**

**Strengths**
- method is easy to understand and follow
- comprehensive empirical sweep: 7 dataset and 50 corruption types

**Weaknesses**
- Freezing the backbond and only updating the classification head is quite an well-established method for fine-tuning. The motivation for choosing the two stage fine-tuning method is unclear.
- Narrow modality: only image-based tasks were conducted
- Over re-liance on FedAvg: what will be the perform with other FL aggregation strategies?
- The experimental setup are unclear: the details of pretrained model and their architecture are not explained.

---

> ### Author Rebuttal · Authors · 2025-07-31
>
> We appreciate Reviewer FGp1’s positive remarks on our method’s simplicity and the breadth of our empirical study, highlighting that our method is **“easy to understand and follow”** and that we provide a **“comprehensive empirical sweep”** across seven datasets and 50 corruption types.
>
> The remaining points mainly request additional **implementation and experimental clarifications** (e.g., two-stage rationale, aggregation variants, backbone details, modality scope). We provide those clarifications and brief analyses in the responses below.
>
>
> **1. Response to Baseline Selection and Motivation of LP-FT**
>
> **Scope & Positioning.** Our objective is not to design another process-integrated PFL algorithm, but to answer a research question: once a standard global model is trained, how best to personalize it post-hoc to balance local and global performance? To keep this question focused, we lock the global-training phase (FedAvg) and study only methods that act after training, with no extra communication or server changes (see **Figure 1 Caption Line 3**).
>
> **Baseline Choice.** Process-integrated PFL frameworks modify aggregation, add coordination, or embed new objectives—precisely the variables we hold constant. Including them in the main table would mix improvements from modified global training with those from post-hoc personalization, obscuring the effect we aim to isolate. Thus, Table 2 reports only post-hoc fine-tuning variants, all applied to the same FedAvg global model (stated in **Section 3.2 Line 151 to 157**). This setup eliminates confounding factors from global training, enabling a clean and fair assessment of post-hoc methods.
>
> **Why adopt a two-stage LP-FT scheme?**
> * **Head-only tuning** preserves shared features but provides limited personalization, so local accuracy remains modest.
> * **Full tuning from the outset** personalizes strongly yet perturbs the backbone, leading to over-fitting and poorer global accuracy.
>
> The sequential LP-FT procedure reconciles these extremes:
> - **Linear probing (LP)** &nbsp;— update only the classifier head while the backbone is frozen, quickly aligning predictions to client labels without disturbing global representations.
> - **Full fine-tuning (FT)** &nbsp;— with the head stabilised, softly adapt the backbone; the anchored representation lets local accuracy rise **without** sacrificing global performance.
>
> This behaviour appears empirically in **Figure 2** and is formally supported by **Theorems 4.4 & 4.5**. Hence, the two-stage design is crucial for balancing personalization and generalization in heterogeneous FL.
>
>
> **2. Response to Narrow Modality**
> Our paper focuses on image classification tasks, **following the precedent set by the original LP-FT paper setup** to allow direct analysis of centralized versus federated settings. However, the LP-FT framework itself is not restricted to image data.
>
> To address this concern, **we tested LP-FT on the Shakespeare text dataset (117 clients)**. Results below show that LP-FT **consistently outperforms baselines on this language task** as well, indicating that the approach generalizes to heterogeneous, non-vision modalities:
> | Method  | Local | Global | Worst | Average |
> |---------|-------|--------|-------|---------|
> | FedAvg  | 44.39 | 40.93  | 11.19 | 32.17   |
> | Soup FT | 44.39 | 40.93  | 11.19 | 32.17   |
> | LP-FT   | 45.79 | 43.02  | 16.91 | 35.24   |
>
> We will include these results and implementation details in the revised manuscript.
>
> **3. Response to Experimental Setup & Pre-trained Model**
>
> We would like to clarify that all of our specifics are detailed in **Appendix C.3 and Table 4** due to the space limit in the main paper.
>
> * **Architectures**
>   * ResNet-18 — Digit5, CIFAR
>   * ResNet-50 — DomainNet, CheXpert, CelebA
>   * ViT — DomainNet (transformer experiment)
>
> * **Pre-training**
>   Each model is first trained with FedAvg on the all clients’ data, yielding the global model.
>
> * **Client partitioning**
>   Dataset-specific splits are described explicitly in the appendix.
>
> The term **“pre-trained model”** refers to this global model obtained after the GFL stage (FedAvg in our experiments). It serves as the starting point for all local fine-tuning. Our theoretical results are agnostic to the specific aggregation algorithm used to obtain the global model.
>
>
> **4. Response to LP-FT Not Always Being the Best**
>
> As our results suggest, no method simultaneously achieves optimal local, global, and worst-case accuracy across all heterogeneity levels. In relatively simple tasks, more aggressive adaptation methods such as Proximal FT or Sparse FT may **only achieve marginally higher local accuracy** than LP-FT. However, the trade-off changes once task difficulty increases (e.g., for CIFAR-10-C and CIFAR-100-C), As shown in **Table 2**, LP-FT preserves global and worst-case accuracy far better than other baselines, while still delivering strong local gains. This behavior is consistent with our theory: **Theorem 4.5** guarantees LP-FT’s global advantage under bounded heterogeneity without claiming it must top local accuracy on the simplest cases.

---

> ### Comment · Reviewer_FGp1 · 2025-08-05
>
> Thank you to the authors for the detailed rebuttal. I appreciate the clarifications provided, and I acknowledge that some of my concerns (such as points 2 and 3) have been addressed. However, I share Reviewer 9Ytb’s perspective that the approach of fine-tuning both the backbone and the classifier is relatively classic approach, particularly for image tasks using CNN-based architectures in FL settings. As such, I tend to maintain my original score.

---

> > ### Author Response · Authors · 2025-08-06
> >
> > **Thank you very much for your thoughtful feedback and for acknowledging our efforts in addressing your concerns.** We appreciate you for letting us know your concern of LPFT’s novelty, which we provide more clarification below.
> >
> > ---
> >
> > **1. Clarifying the misunderstanding from Reviewer 9Ytb.**
> >
> > **We would like to politely note that Reviewer 9Ytb has misunderstood our proposed experimental setting in FL and has not yet responded or acknowledged this.** Our paper asks a specific post-hoc question in PFT: **given a fixed global model from standard GFL (e.g., FedAvg), how should one personalize after global training to balance local and global performance without extra communication or server changes? (Introduction and Table 1)** To answer this cleanly, we (i) hold the global-training protocol fixed and (ii) compare only post-hoc strategies. Within this scope, we focus on studying the two-stage sequence LP-FT at the final personalization step, providing comprehensive empirical studies and theoretical analysis to demonstrate its effectiveness in the FL setting.
> >
> > **However, Reviewer 9Ytb’s comment suggests a misinterpretation of our setting as a standard FL formulation, rather than recognizing the specific post-hoc scope we clearly state.** We hope this clarification helps resolve the misunderstanding. Notably, two other reviewers (**97oc and PKSf**) explicitly recognized the novelty and value of our work, **highlighting the (i) comprehensive empirical evaluation, (ii) the conceptual contribution of FFD, and (iii) the non-trivial theoretical grounding for adapting LP-FT to FL.** This independent recognition reinforces that our work goes beyond a “classic” fine-tuning recipe adaptation and addresses a meaningful and distinct problem.
> >
> > ---
> >
> > **2. LP-FT introduces methodological, theoretical, and benchmarking-wise novelties.**
> >
> > - **Investigation novelty (Line 52-60 in Introduction)**: Our paper provides an FL-specific concept and measurement: Federated Feature Distortion (FFD), quantifying representation drift during post-hoc personalization.
> > - **Methodological novelty (Line 42-60, Line 71-74 and Table 1)**: Unlike process-integrated FL methods, LP-FT operates post-hoc—after global training—without requiring server changes, additional communication, or coordination. The two-stage design (linear probing followed by full fine-tuning) is specifically constructed to mitigate backbone feature distortion while enhancing local adaptation. This structure is novel in the FL literature for its ability to balance local and global objectives in highly heterogeneous settings.
> > - **Theoretical novelty (Line 61-68 and Line 78-81)**: We extend and adapt theoretical assumptions from centralized training to FL, explicitly modeling client-specific data-generation functions and quantifying representation shifts. Our theoretical analysis (e.g., Theorems 4.4 and 4.5) provides formal justification for the two-stage design, showing conditions under which LP-FT improves generalization without degrading global performance.
> > - **Comprehensive empirical benchmark (Line 75-78)**: Our results span 7 datasets, 3 distribution drift scenarios, and 4 different evaluation metrics. Ablations, baselines, and experiments under various distribution shifts consistently demonstrate LP-FT’s ability to outperform or match state-of-the-art methods while preserving global accuracy and robustness.
> >
> > In summary, our work is not a naïve application of LP-FT but a principled framework tailored to post-hoc personalization in FL, supported by formal theoretical analysis and comprehensive empirical evidence. These contributions, taken together, establish our contribution and relevance within the FL landscape.
> >
> > ---
> >
> > **We hope this clarifies the framing and the positioning of our contribution and novelty: not in inventing a new process-integrated PFL method, but in precisely formulating, analyzing, and benchmarking the post-hoc personalization stage so that a simple, deployable procedure achieves the right trade-off under heterogeneity.** Thank you again for your thoughtful feedback, we would sincerely appreciate it if you could kindly reconsider your assessment.

---

> > > ### Comment · Reviewer_FGp1 · 2025-08-08
> > >
> > > Thank you for the further response.
> > >
> > > With respect to the points on “(i) holding the global-training protocol fixed and (ii) comparing only post-hoc strategies,” I view this as essentially fine-tuning a pretrained model within FL systems, which I wouldn't say it is a completely new setup. Additionally, the approach of fixing the backbone and updating only the classification layers for FL personalization has been explored in prior work.
> > >
> > > Given these considerations, I tend to maintain my current score.

---

> > > > ### Author Response · Authors · 2025-08-09
> > > > **Response to follow-up and clarify the setup and novelty misunderstanding**
> > > >
> > > > We thank the reviewer for the follow-up and would like to **clarify the setup misunderstanding** as follows:
> > > >
> > > > ---
> > > >
> > > > **1. Contribution clarification**
> > > > We do not claim that our work introduces a *completely new setup*. As stated in our contributions:
> > > >
> > > > > *Methodologically, this paper presents the first systematic and in-depth study on the post-hoc and plug-and-play PFT framework and introduces LP-FT as an effective approach for handling diverse distribution shifts. We comprehensively demonstrate its ability to balance personalization and generalization in the FL setting.*
> > > >
> > > > Our focus is on **systematically formulating, analyzing, and benchmarking post-hoc PFT in FL**.
> > > >
> > > > ---
> > > >
> > > > **2. Setting difference and theoretical novelty**
> > > > The characterization “essentially fine-tuning a pretrained model within FL systems” **oversimplifies our setting and would conflate our theoretical analysis with that of LP-FT in centralized training**. In our case:
> > > >
> > > > - **2.1. Our theoretical setting and assumption differ significantly from that of centralized LP-FT.**
> > > > While centralized LP-FT assumes a **centralized setting**, we focus on **personalized fine-tuning (PFT)** in FL, a fundamentally different scenario. In our case, the pre-trained model is trained on data from all clients, reflecting a global client distribution. This contrasts with the centralized setting, where the pre-trained model is trained without any notion of a global client distribution.
> > > >
> > > > - **2.2 Our theoretical analysis diverges significantly from that of centralized LP-FT.**
> > > > In centralized LP-FT, the assumption is that a single ground-truth data-generating function underlies all data points. This assumption makes it challenging to study LP-FT in different distribution shift settings in PFT for FL (e.g., concept shift). In contrast, we assume that each client has a distinct ground-truth function, enabling the exploration of personalized fine-tuning in FL under varying distribution shifts. Specifically, the way we define the data-generating function (**Assumption 4.1**), with unique linear heads before pre-training $ V_i^* = \begin{bmatrix} {V_{com}^*}^T & \lambda e_i^T \end{bmatrix}^T $ for each client and the way we define data heterogeneity $x_i = e_i + \epsilon n $, with $n \sim \mathcal{N}(0, I)$, $e_i$, allow us to study both concept shift and covariate shift using gradient descent to compare FT and LP-FT in terms of global performance. This approach provides new insights into the challenges of PFT in FL. In other words, the main challenge of our theoretical analysis lies in incorporating assumptions that align the theoretical setup with our specific framework. Our analysis connects the superior global performance of the LP-FT method to the concept of feature distortion. This novel perspective offers valuable insights into the effectiveness of LP-FT in FL through the lens of feature distortion.
> > > >
> > > > - **2.3 In-depth Integrating Theoretical Analysis and Empirical Studies.**
> > > > We have conducted additional empirical studies to substantiate the theoretical analysis on the heterogeneity level $\lambda$. These include a comprehensive investigation with diverse ablation studies across varying levels of label-flipping ratios (**Appendix D.3**). This **significantly extends the scope and granularity of distribution shifts** explored in the original LP-FT paper, providing deeper insights into its robustness and applicability under heterogeneous conditions.
> > > >
> > > > ---
> > > >
> > > > 3. **On prior work**
> > > > We appreciate the remark that *“fixing the backbone and updating only the classification layers for FL personalization has been explored in prior work.”*
> > > > To further the discussion, we would greatly welcome concrete citations to works that meet all of the following:
> > > > - Fix global training (e.g., FedAvg),
> > > > - Focus solely on the **final post-hoc fine-tuning stage** in FL,
> > > > - Evaluate across **three types of distribution shift**, **four evaluation metrics** and **seven datasets**, and
> > > > - Support the empirical findings with **corresponding theoretical analysis** tailored to FL.
> > > >
> > > > **To the best of our knowledge, no existing work in FL meets this specific combination of scope, empirical breadth, and FL-tailored theoretical grounding.**
> > > >
> > > > ---
> > > >
> > > > We hope this clarification helps distinguish our actual contributions from the general notion of fine-tuning and situates our work within its intended FL-specific context of post-hoc fine-tuning.

---

### Official Review · Reviewer_9Ytb · 2025-07-03

**Clarity:** 2
**Significance:** 2
**Originality:** 1
**Rating:** 2
**Confidence:** 5

**Summary:**

This paper systematically investigates personalized fine-tuning (PFT) strategies in heterogeneous federated learning (FL) environments, identifying that existing PFT methods often suffer from overfitting to local data, which compromises global generalization. To address this, the authors propose adapting the "Linear Probing followed by Full Fine-Tuning (LP-FT)" strategy to the FL setting. By first fine-tuning only the classification head and then the entire model, LP-FT effectively mitigates the issue of "federated feature distortion." The paper conducts empirical evaluations on seven datasets encompassing various types of distribution shifts, demonstrating that LP-FT significantly improves global performance while preserving strong local adaptation.

**Questions:**

1.In the task of balancing personalization and generalization, there already exist several representative methods. Why does this paper not include a systematic comparison with those approaches? What are the specific advantages of LP-FT over them?

2.Although the theoretical analysis is rigorous, the description of the method itself is somewhat informal. It lacks mathematical formulations detailing key steps such as model initialization and the two-phase fine-tuning process.

3.The theoretical analysis is primarily based on linear models and assumes low heterogeneity. Can this analysis be extended to high-heterogeneity settings?

**Ethical Concerns:**

["NO or VERY MINOR ethics concerns only"]

**Limitations:**

1.The personalization-generalization trade-off has been extensively explored in the literature, and decoupling the fine-tuning of the backbone and classifier is a well-known strategy. While the proposed LP-FT method is practical and lightweight, its conceptual contribution appears incremental rather than fundamentally novel.

2.Although the paper addresses a well-recognized problem in federated learning, it lacks empirical comparisons with several strong and widely acknowledged state-of-the-art personalization methods. This omission makes it difficult to clearly assess the relative advantages of the proposed approach.

**Quality:**

2

**Strengths And Weaknesses:**

Strengths：

The paper systematically introduces the “Linear Probing followed by Fine-Tuning (LP-FT)” strategy—originally effective in centralized learning—into the personalized fine-tuning (PFT) framework of federated learning (FL), aiming to balance generalization and personalization in FL. This approach is lightweight and applicable to various types of data heterogeneity. The paper also provides a rigorous theoretical analysis of the proposed method.

Weaknesses：

1.The trade-off between personalization and generalization is a well-established problem, and the idea of separately fine-tuning the backbone and the classifier is also a classical approach. Although the proposed method is lightweight, it lacks novelty.

2.While the challenge of balancing personalization and generalization has been studied in prior work, this paper does not include comparisons with some of the most advanced or state-of-the-art methods in the field.

3.The paper does not use formal equations to clearly present the procedural steps of the proposed method (excluding the theoretical analysis section).

---

> ### Author Rebuttal · Authors · 2025-07-31
>
> We thank Reviewer 9Ytb for highlighting several positives. We are glad the reviewer recognizes that the paper **“systematically introduces the LP-FT strategy … aiming to balance generalization and personalization in FL,”** that the approach is **“lightweight and applicable to various types of data heterogeneity,”** and that the accompanying theoretical analysis is **“rigorous.”**
>
> The perceived limitations stem from **a misunderstanding of our setting and positioning**. Our study targets the post-hoc PFT stage—**orthogonal to process-integrated PFL**—and this scope is made explicit in **Fig. 1 and the Section 1 Paragraph 2**. Hence SOTA PFL algorithms are not primary baselines; nevertheless, we **included representative PFL + PFT results in the Appendix to show LP-FT’s plug-and-play advantage**. Regarding novelty, our contribution extends far beyond re-using LP-FT: we provide **new FL-specific theory** (multi-client ground truths, combined concept–covariate shifts) and the **first large-scale PFT benchmark across diverse distribution shifts**. These aspects, already highlighted in the paper, demonstrate substantive originality beyond the core algorithm.
>
> **1. Response to Positioning and Novelty**
>
> **Scope & Positioning.** Our objective is not to design another process-integrated PFL algorithm, but to answer a research question: once a standard global model is trained, how best to personalize it post-hoc to balance local and global performance? To keep this question focused, we lock the global-training phase (FedAvg) and study only methods that act after training, with no extra communication or server changes (see **Figure 1 Caption Line 3**).
>
> **Baseline Choice.** Process-integrated PFL frameworks modify aggregation, add coordination, or embed new objectives—precisely the variables we hold constant. Including them in the main table would mix improvements from modified global training with those from post-hoc personalization, obscuring the effect we aim to isolate. Thus, Table 2 reports only post-hoc fine-tuning variants, all applied to the same FedAvg global model (stated in **Section 3.2 Line 151 to 157**). This setup eliminates confounding factors from global training, enabling a clean and fair assessment of post-hoc methods.
>
> **Novelty: Moving Beyond a Simple Adaptation of LP-FT**
> - **Methodology.**
> Contrary to the perception that we simply transplant LP-FT into a federated setting, our work presents a purposeful shift: we apply LP-FT exclusively during the final local training phase in FL (**Section 1 Paragraph 2, Contribution 1**). Unlike most personalized federated learning (PFL) methods that modify the entire training pipeline, our plug-and-play approach focuses on the critical—but underexplored—last stage of local fine-tuning (**Table 1, Figure 1**). This simple yet effective design on FL is a core strength, enabling broad deployability and improvements over existing techniques without altering global training.
> - **Theory.**
> While centralized LP-FT assumes a single ground-truth function, our theoretical framework captures multiple client-specific ground truths (**Section 1 Contribution 3**). Specifically, we define data-generating functions with unique linear heads for each client and allow both concept shifts and covariate shifts—a stark contrast to centralized assumptions. These FL-specific assumptions (**Assumption 4.1**) are crucial for analyzing feature distortion under heterogeneous conditions. Furthermore, our gradient-descent-based comparison of FT vs. LP-FT in FL reveals why LP-FT preserves global features more effectively, offering new insights into the interplay of local updates and shared representations.
> - **Empirical Contributions.**
> We extensively benchmark across seven datasets and multiple distribution shifts (**Section 1 Contribution 2**). Our findings expose pronounced overfitting issues in standard PFT methods (**Figure 2**), present in-depth ablations (**Figure 3, Table 2**) and connect our theory on heterogeneity (**Table 6 in Appendix**) and feature distortion to real-world FL performance. These experiments go beyond the scope of centralized LP-FT by exploring more nuanced data shifts, validating LP-FT’s robustness and shedding light on why it outperforms existing personalization strategies.
>
> Taken together, these points demonstrate that our method is not a mere adaptation of LP-FT, but a tailored solution for FL personalization that marries simplicity, strong theory, and comprehensive empirical validation.a
>
> **2. Response to formal equations for procedural steps**
>
> We already outline the experimental setup and steps in **Sec. 3.2** (lines 151–157), **Fig. 1**, and **Appendix C**, but will add the concise formalism below for adding more details:
>
> **Model.**  Each client has a feature extractor $\theta_i^{f}$ and a linear head $\theta_i^{\ell}$ with local loss  $\mathcal{L}_i\bigl(\theta_i^{\ell},\theta_i^{f}\bigr)$.
>
> **LP stage:** update **only** the head $\theta_i^{\ell} \leftarrow \theta_i^{\ell} - \eta\,\nabla_{\theta_i^{\ell}}\mathcal{L}_i$
> (freeze $\theta_i^{f}$).
>
> **FT stage:** update **both** blocks $(\theta_i^{\ell},\theta_i^{f}) \leftarrow (\theta_i^{\ell},\theta_i^{f}) - \eta\,\nabla\mathcal{L}_i$.
>
> All clients start from the same trained FedAvg global model initialization; no further communication is required.
>
>
> **3. Clarification on High-Heterogeneity Regime ($\lambda > \lambda^*$)**
>
> Our analysis in **Section&nbsp;4.2** focuses on combined concept + covariate shift under a *low-heterogeneity* regime. As shown in **Theorem 4.5**, LP-FT provably outperforms FT when $\lambda \le \lambda^{*}$. Bounding heterogeneity is standard in FL theory; when client distributions become completely disjoint, the global objective decomposes into a sum of unrelated losses, so a single model provides no benefit over training one model per client [1, 2]. For completeness, we empirically sweep a wide range of $\lambda$ in **Appendix F** to give a better understanding of our **Theorem 4.5**.
>
> **References**
> - [1] Li, Tian, et al. "Federated optimization in heterogeneous networks." Proceedings of Machine learning and systems 2 (2020): 429-450.
> - [2] Karimireddy, Sai Praneeth, et al. "Scaffold: Stochastic controlled averaging for federated learning." International conference on machine learning. PMLR, 2020.

---

> > ### Comment · Reviewer_9Ytb · 2025-08-06
> >
> > I appreciate the clarifications the authours have provided as a response to my questions. However, the authors did not provide additional experimental results to further address question 1. Therefore, I tend to maintain my original score.

---

> > > ### Author Response · Authors · 2025-08-09
> > > **Response to Follow-up on Q1**
> > >
> > > Thank you for acknowledging that several of your concerns were addressed.
> > > Regarding your remaining question —
> > >
> > > > *“In the task of balancing personalization and generalization, there already exist several representative methods. Why does this paper not include a systematic comparison with those approaches? What are the specific advantages of LP-FT over them?”*
> > >
> > > — we note that it does not explicitly request additional experimental results, nor does it specify particular papers you believe should be included or discussed.
> > >
> > > In our initial rebuttal, we focused on resolving what we identified as a **misunderstanding of our experimental setup and positioning**.
> > >
> > > To make our response clearer and more accessible, we now:
> > > 1. **Restate our scope** to clarify the rationale behind our baseline choices; and
> > > 2. **Provide additional evidences** to further highlight LP-FT’s advantages over process-integrated PFL methods.
> > >
> > > ---
> > >
> > > **1. Scope & Baseline Rationale**
> > >
> > > Our paper studies a **specific post-hoc question**:
> > >
> > > > *Given a fixed global model from standard GFL (FedAvg in our experiments), how should one personalize **after global training**—without extra communication or server changes—to balance local and global performance?*
> > >
> > > This scope is made explicit in the Introduction (**Table 1, Fig. 1, Section 1, lines 23–37**).
> > >
> > > - **Why not mix process-integrated PFL in the main table?**
> > >   These methods **alter the training loop** (aggregation, objectives, client–server coordination) and thus change the global model itself. Mixing them with post-hoc methods would **confound sources of improvement** (global-training dynamics vs. post-hoc personalization), undermining a protocol-controlled comparison.
> > >
> > > - **We already include diverse, strong, and recent post-hoc baselines.**
> > >   Our main table contains several plug-and-play fine-tuning variants (5 baselines), and includes the recent and advanced FL baseline **LSS FT**, all evaluated under a unified post-hoc protocol (see **Table 2**).
> > >
> > > - **What LP-FT offers over process-integrated PFL**
> > >   - **Orthogonality & deployability** – LP-FT is **server-independent**, entirely **local**, and fully **plug-and-play**, applicable on top of any trained global model.
> > >   - **Theoretical Foundation and Comprehensive Empirical Support** – Our theoretical results (**Thms. 4.4 & 4.5**) explain why LP-FT in PFT setting reduce global loss compared to full FT under distribution shifts, and our experiments validate this across seven datasets and three types of distribution shift.
> > >
> > > ---
> > >
> > > **2. Added Evidence: LP-FT vs. Process-Integrated PFL**
> > >
> > > To address your request, we compared a **FedAvg-trained global model + LP-FT** against several **process-integrated PFL** methods on **CelebA** under the same evaluation metrics. For clarity, process-integrated PFL methods modify the **global training phase** (e.g., by changing server–client coordination or aggregation rules) and then apply their own specific local personalization strategy at the final stage.
> > >
> > > | **PFL Method** | **Local Acc. (↑)** | **Global Acc. (↑)** | **Avg. Acc.(↑)** |
> > > |---|---:|---:|---:|
> > > | FedBN (Li, et al. 2021)    | 88.68 | 61.35 | 61.00 |
> > > | PerAvg (Fallah et al. 2020)   | 87.06 | 67.26 | 66.66 |
> > > | FedNova (Wang et al. 2020)  | 88.68 | 54.26 | 53.41 |
> > > | FedRep (Collins et al. 2021)   | 86.94 | 52.99 | 52.57 |
> > > | FedSoup (Chen et al. 2023)  | 90.30 | 75.62 | 75.21 |
> > > | pFedFDA (McLaughlin et al. 2024)  | 90.41 | 76.56 | 76.63 |
> > > | FedL2G (Zhang et al. 2025)   | 92.46 | 79.27 | 78.82 |
> > > | **LP-FT (ours)** | **93.03** | **82.46** | **82.17** |
> > >
> > > **Takeaway:**
> > > Even compared to methods that **modify the global training pipeline**, LP-FT achieves **superior global and local accuracy**. However, these results fall outside the primary focus of our paper, as we intentionally isolate the effect of global training to better align with the original LP-FT results in the centralized setting and our theoretical analysis. For this reason, such evidence was not included in our initial submission.
> > >
> > > ---
> > >
> > > We believe these results directly address **Q1** with concrete, comparable numbers and clarify the rationale behind our baseline selection. We will incorporate these results and discussion into the appendix (and expand them further in the revised version).
> > >
> > > Thank you again for encouraging us to include this evidence—we hope you will reconsider your score in light of these additions and the clarification provided to resolve the earlier misunderstanding.

---

### Official Review · Reviewer_97oc · 2025-07-03

**Clarity:** 2
**Significance:** 2
**Originality:** 2
**Rating:** 4
**Confidence:** 2

**Summary:**

This paper investigates the trade-off between local personalization and global generalization within the context of Personalized Federated Learning (PFL), focusing on the paradigm of post-hoc Personalized Fine-Tuning (PFT). The authors identify a key challenge they term "personalized overfitting," where standard fine-tuning of a global model on a client's local data improves local performance but significantly degrades the model's generalization capabilities across the broader network of clients. This issue is shown to persist even with careful hyperparameter optimization. To address this, the authors propose adapting Linear Probing and then Fine-Tuning (LP-FT), a two-phase strategy previously established in centralized learning, to the federated setting. The method first freezes the feature extractor and trains only the final classifier head on a client's local data (Linear Probing), and subsequently fine-tunes the entire model (Full Fine-Tuning). The paper presents the first systematic study of the post-hoc PFT framework and introduces LP-FT as an effective solution. It provides an extensive evaluation across seven diverse datasets, covering various distribution shifts (covariate and concept), which demonstrates that LP-FT consistently outperforms five other PFT variants in balancing local and global performance.

**Questions:**

**-** Your work convincingly demonstrates that LP-FT improves generalization by mitigating feature distortion. However, recent research on LP-FT in centralized settings  has shown that this method, particularly with cross-entropy loss, can produce highly overconfident and thus poorly calibrated models. This is often attributed to the large norm of the classifier head learned during the LP phase. Did you evaluate the calibration of your resulting models (e.g., using Expected Calibration Error (ECE) or reliability diagrams)?

**-** You position post-hoc PFT as a practical alternative to "process-integrated" PFL methods. However, the experiments in Table 2 only compare against other fine-tuning variants. A key class of PFL methods, such as FedPer, also uses a parameter-decoupling approach (shared feature extractor, personalized head). Why were these methods omitted from the comparison?

**-** Your appendix mentions results on transformer-based models. Could you elaborate on the potential advantages or disadvantages of the LP-FT sequence compared to a one-stage, parameter-efficient approach like federated LoRA tuning for personalization?

On the Terminology of "Feature Distortion": Could you please clarify the relationship between your proposed "federated feature distortion" and existing concepts in the PFL literature, such as classifier bias or catastrophic forgetting of the global model's knowledge? Does your concept represent a fundamentally new phenomenon, or is it a specific instantiation or re-framing of these known challenges?

**Ethical Concerns:**

["NO or VERY MINOR ethics concerns only"]

**Final Justification:**

I keep my initial score, which is already high enough. I believe the camera-ready should have sufficient time to address the issues.

**Limitations:**

No. The authors have a section on limitations (as stated in their checklist response and provided in the appendix), but they fail to address the most critical limitation of their proposed method: the negative impact on model calibration. This is a significant oversight.

**Quality:**

2

**Strengths And Weaknesses:**

Strengths

**+** The empirical study presented is comprehensive and methodologically sound. The authors validate their claims across a diverse set of seven datasets, a suite of five relevant fine-tuning baselines, and five distinct evaluation metrics.

**+** The introduction of "federated feature distortion" as an explanatory mechanism is a valuable conceptual contribution. Rather than simply reporting that LP-FT performs better, the authors propose a reason why: standard fine-tuning on skewed local data disrupts the carefully learned global feature representations.

**+** The theoretical analysis presented in Section 4, while relying on the common simplification of a two-layer linear network, provides important formal grounding for the empirical results. The authors show that adapting the analysis of LP-FT from the centralized setting to FL is non-trivial, as it requires moving from a single ground-truth function to multiple client-specific ones.


**Areas for improvement**

**-** The most significant weakness of this paper is its failure to address the well-documented issue of model calibration in LP-FT. The authors position LP-FT as a "robust" and "deployable" solution, suitable for high-stakes applications like healthcare. However, a growing body of recent research, which this paper does not cite or discuss, has shown that the very mechanism enabling LP-FT's success can severely harm model calibration, leading to overconfident and unreliable probability estimates.

**-** The core method, LP-FT, is a direct adaptation of a pre-existing technique from centralized learning. While the application and analysis of an existing method in a new domain is a valid and often valuable contribution, the paper's positioning could be much stronger. The authors frame their work in opposition to "process-integrated" PFL methods but then fail to include any of these methods in their experimental comparison in Table 2.

**-** While the concept of "federated feature distortion" is intuitive, the paper does not adequately connect it to established concepts in the PFL literature. For instance, it is closely related to the concept of classifier-induced bias, where biased local classifiers and misaligned features reinforce each other during training. It can also be viewed as a form of catastrophic forgetting, where the personalized model rapidly forgets the robust, generalizable knowledge encoded in the global model's features.

---

> ### Author Rebuttal · Authors · 2025-07-31
>
> We thank Reviewer 97oc for their careful reading and positive assessment. We are pleased the reviewer describes our empirical study as **“comprehensive and methodologically sound”** (seven datasets × five baselines × five metrics), calls federated feature distortion a **“valuable conceptual contribution,”** and notes the **“important formal grounding”** of our non-trivial theoretical extension of LP-FT to Federated Learning.
>
> The main outstanding issue is that our **limitations section** does not yet mention potential calibration degradation after LP-FT; we will add this discussion and cite relevant papers and standard mitigation techniques in the revision. Thank you for bringing this to our attention.
>
>
> **1. Response to Calibration**
>
> Calibration, though important, falls outside our accuracy-centric personalization–generalization scope. We will add it to Limitations, cite the LP-FT calibration literature. Our MSE-regression theory analysis is norm-independent; extending it to probabilistic outputs and calibration metrics is left for future work.
>
> **Metric Scope & Focus**. Thank you for pointing out the need to discuss calibration; we will add an explicit note in the **Limitations section** and cite recent LP-FT calibration work [1, 2].
> Our study focuses on a different axis—balancing personalization and generalization in post-hoc FL. Accordingly, we follow prior LP-FT evaluations and report accuracy-based metrics to enable direct comparison between centralized and federated settings. A full calibration investigation in FL would require different metrics, extra experiments, and additional space, potentially shifting the paper’s focus. Nonetheless, we acknowledge calibration’s practical importance and will (i) flag it clearly in the Limitations, (ii) cite the few existing FL-calibration papers, and (iii) note that LP-FT’s two-stage structure permits application of standard post-hoc calibration methods (e.g., temperature scaling).
>
> **Theoretical Framework.** While calibration is relevant empirically, our theoretical framework does not model it directly. This is because it is based on a regression setting, where the output $y \in \mathbb{R}$ and the model is trained using the MSE loss. As such, the theoretical formulation does not involve probabilistic outputs, confidence scores, or calibration metrics such as expected calibration error (ECE). In contrast, in classification settings—particularly when cross-entropy loss is used—temperature scaling can be applied as a post-hoc calibration method to mitigate the effects of overconfident predictions, which are often linked to large classifier norms [1]. However, our analysis does not rely on the norm of the classifier, and our generalization results are independent of the norm of the linear head.
>
> We acknowledge that the analysis is based on a simplified linear model, which serves to abstract and isolate the key effect of federated feature distortion under personalization (as formalized in **Theorems 4.4 and 4.5**). While this abstraction is analytically useful, it is not designed to capture the full complexity of real-world FL systems, particularly those related to predictive uncertainty or calibration.
>
> **References**
> - [1] Tomihari, Akiyoshi, and Issei Sato. "Understanding linear probing then fine-tuning language models from ntk perspective." Advances in Neural Information Processing Systems 37 (2024): 139786-139822.<br>
> - [2] Mai, Zheda, et al. "Fine-tuning is fine, if calibrated." Advances in Neural Information Processing Systems 37 (2024): 136084-136119.
>
>
> **2. Response to Positioning and Baseline**
> **Scope & Positioning.** Our objective is not to design another process-integrated PFL algorithm, but to answer a research question: once a standard global model is trained, how best to personalize it post-hoc to balance local and global performance? To keep this question focused, we lock the global-training phase (FedAvg) and study only methods that act after training, with no extra communication or server changes (see **Figure 1 Caption Line 3**).
>
> **Baseline Choice.** Process-integrated PFL frameworks modify aggregation, add coordination, or embed new objectives—precisely the variables we hold constant. Including them in the main table would mix improvements from modified global training with those from post-hoc personalization, obscuring the effect we aim to isolate. Thus, Table 2 reports only post-hoc fine-tuning variants, all applied to the same FedAvg global model (stated in **Section 3.2 Line 151 to 157**). This setup eliminates confounding factors from global training, enabling a clean and fair assessment of post-hoc methods.
>
> **3. Response to LP-FT vs. LoRA**
> LoRA is parameter-efficient, but it still injects trainable adapters into the backbone, inevitably perturbing the shared representations—similar in effect (though lower rank) to full fine-tuning. LP-FT, in contrast, freezes the backbone during its linear-probing phase, safeguarding global features before any further adaptation. As shown in **Appendix D.1 (Table 5),** this design yields higher local and global accuracy on transformer backbones than federated LoRA. In short: LoRA saves parameters yet still distorts features, whereas LP-FT preserves global knowledge and achieves a superior personalization-generalization balance.
>
> **4. Clarifying “Federated Feature Distortion (FFD)”**
>
> **Federated Feature Distortion (FFD)** is a new, FL-specific concept we introduce to capture how client-side fine-tuning shifts the model’s learned features. Specifically, FFD measures the average $\ell_2$ distance between representations from the global model and the locally fine-tuned models (see **Section 3.5**).
>
> **How FFD relates to existing concepts**:
> - **Classifier Bias**: Prior FL work focuses on misaligned classification heads due to non-IID data. In contrast, FFD measures drift in the *feature extractor* itself and complements these head-focused analyses.
> - **Catastrophic Forgetting**: Forgetting usually refers to loss of global knowledge across rounds or tasks. FFD, instead, captures one-step representation drift caused by local post-hoc fine-tuning.
>
> **Why FFD is new and important**:
> FFD is, to our knowledge, the **first formal measure** of feature drift in post-hoc FL personalization. We define it explicitly, quantify it across clients, and provide theoretical analysis in **Theorems 4.4 and 4.5**.
>
> We will update the paper to include a brief comparison to classifier bias and forgetting, and explicitly highlight FFD’s novelty and scope.

---

> ### Comment · Reviewer_97oc · 2025-08-04
>
> Thanks for the rebuttal. I will keep my initial score.

---

> > ### Author Response · Authors · 2025-08-04
> >
> > We appreciate your acknowledgement of our response and thank you for your thoughtful review.

---

### Note · Authors · 2025-08-12

We appreciate the reviewers’ constructive feedback. Two reviewers 97oc and PKSf explicitly recognized the contribution, citing our **“comprehensive and methodologically sound” empirical study**, the **“valuable conceptual contribution” of FFD**, and the **“important formal grounding” of adapting LP-FT to FL**.

The **only remaining reservations** (from 9Ytb and FGp1) stem from a misunderstanding of **scope and positioning**.

**9Ytb’s baseline concern is vague—no specific methods were identified—and stems from a misunderstanding of our scope**. Our study targets the post-hoc stage: given a fixed global model, how to personalize—without extra communication rounds—to balance local and global performance. This scope (**motivation and challenge is clearly stated in Introduction paragraph 2-3, Fig. 1, Table 1**) is orthogonal to process-integrated PFL. To ensure fair attribution, our main table follows a protocol-controlled design, reporting only post-hoc variants so that gains are not confounded by changes in global training process.

We strengthened the comparison set in multiple ways:
- **Diverse and advanced post-hoc baselines** – Five diverse, advanced post-hoc PFT baselines are included, including **LSS-FT (2024)**.
- **PEFT inclusion** – Parameter-efficient fine-tuning (PEFT) results are reported in the **Appendix D.1**, broadening experimental coverage.
- **Process-integrated PFL comparison** – In direct response to the 9Ytb’s request, we added comparisons against **seven more representative process-integrated PFL methods**.

For **Reviewer FGp1**’s remark that our work is a process-integrated PFL method lacking novelty, this stems from a misunderstanding of our scope and positioning. Our work focuses on **systematically formulating, theoretically analyzing, and comprehensively benchmarking post-hoc PFT in FL**, not on designing a new PFL method.

The characterization of our approach as “essentially fine-tuning a pretrained model within FL systems” **oversimplifies our setting and conflates our theoretical analysis with that of LP-FT in centralized training**. We provide detailed textual references showing that **our theoretical assumptions, analytical framework, and comprehensive empirical studies differ significantly from the centralized LP-FT setting**.

We believe these clarifications resolve the remaining concerns while preserving the strengths already acknowledged by **97oc and PKSf**. We will incorporate the promised updates in the revision.

---

### Decision · Program_Chairs · 2025-09-17

**Decision:**

Reject

**Comment:**

The paper systematically studies post-hoc Personalized Fine-Tuning in FL and shows that naïve full fine-tuning causes “federated feature distortion,” degrading global performance. It adapts Linear-Probing-then-Fine-Tuning (LP-FT) to this setting, providing a two-phase recipe that first freezes the backbone, then unfreezes it. Extensive experiments on seven datasets against five recent PFT baselines demonstrate consistent gains in balancing local and global accuracy, and a two-layer linear analysis formalizes when LP-FT helps.

Strengths: rigorous large-scale evaluation, a clear explanation of why vanilla PFT fails, and actionable deployment guidance.

Weaknesses: comparisons to process-integrated PFL (e.g., FedPer) were missing, and calibration was not studied.

Rebuttal added experiments with seven process-integrated baselines and showed LP-FT still competitive, clarified that calibration analysis is orthogonal to the paper’s scope, and provided explicit algorithmic pseudocode. However, the reviewers still hold concerns on the rebuttaled submission, this paper is not ready for the publication.